# Transport Properties of Methyl-Terminated Germanane Microcrystallites

**DOI:** 10.3390/nano12071128

**Published:** 2022-03-29

**Authors:** Davide Sciacca, Maxime Berthe, Bradley J. Ryan, Nemanja Peric, Dominique Deresmes, Louis Biadala, Christophe Boyaval, Ahmed Addad, Ophélie Lancry, Raghda Makarem, Sébastien Legendre, Didier Hocrelle, Matthew G. Panthani, Geoffroy Prévot, Emmanuel Lhuillier, Pascale Diener, Bruno Grandidier

**Affiliations:** 1UMR 8520-IEMN, Université de Lille, CNRS, Centrale Lille, Université Polytechnique Hauts-de-France, Junia-ISEN, 59000 Lille, France; daveshake93@gmail.com (D.S.); maxime.berthe@iemn.fr (M.B.); nemanja.peric@iemn.fr (N.P.); dominique.deresmes@iemn.fr (D.D.); louis.biadala@iemn.fr (L.B.); christophe.boyaval@iemn.fr (C.B.); pascale.diener@junia.com (P.D.); 2Department of Chemical and Biological Engineering, Iowa State University, Ames, IA 50011, USA; bryan@iastate.edu (B.J.R.); panthani@iastate.edu (M.G.P.); 3UMR 8207–UMET-Unité Matériaux et Transformations, Université de Lille, CNRS, INRAE, Centrale Lille, 59000 Lille, France; ahmed.addad@univ-lille.fr; 4HORIBA FRANCE SAS, 91120 Palaiseau, France; ophelie.lancry@horiba.com (O.L.); raghda.makarem@horiba.com (R.M.); sebastien.legendre@horiba.com (S.L.); didier.hocrelle@horiba.com (D.H.); 5Institut des NanoSciences de Paris, CNRS, Université de Sorbonne, 75005 Paris, France; geoffroy.prevot@insp.jussieu.fr (G.P.); lhuillier@insp.jussieu.fr (E.L.)

**Keywords:** germanane, methylation, hydration, resistivity, thermal robustness

## Abstract

Germanane is a two-dimensional material consisting of stacks of atomically thin germanium sheets. It’s easy and low-cost synthesis holds promise for the development of atomic-scale devices. However, to become an electronic-grade material, high-quality layered crystals with good chemical purity and stability are needed. To this end, we studied the electrical transport of annealed methyl-terminated germanane microcrystallites in both high vacuum and ultrahigh vacuum. Scanning electron microscopy of crystallites revealed two types of behavior which arise from the difference in the crystallite chemistry. While some crystallites are hydrated and oxidized, preventing the formation of good electrical contact, the four-point resistance of oxygen-free crystallites was measured with multiple tips scanning tunneling microscopy, yielding a bulk transport with resistivity smaller than 1 Ω·cm. When normalized by the crystallite thickness, the resistance compares well with the resistance of hydrogen-passivated germanane flakes found in the literature. Along with the high purity of the crystallites, a thermal stability of the resistance at 280 °C makes methyl-terminated germanane suitable for complementary metal oxide semiconductor back-end-of-line processes.

## 1. Introduction

The aromatic bonds of graphene can be saturated with hydrogen atoms. This process leads to a 2D hydrocarbon called graphane [1,2], where the flat morphology of graphene evolves in a buckled sheet. This symmetry breaking results in a band gap opening to a value between 3.5 and 3.7 eV, depending on the resulting configuration of the buckling (boat or chair/twist) [3,4]. Such a wide band gap hampers the use of graphane in electronics and has motivated the synthesis of silicon and germanium-based analogues of graphane, called silicane and germanane, respectively [5,6,7]. Both silicane and germanane have a reduced theoretical band gap compared to graphane: 2.9 eV and 1.9 eV, respectively [8]. The experimental value has been found to be even lower, between 1.4 and 1.6 eV for germanane [6,9], with carrier mobilities of tens of cm^2^·V^−1^·s^−1^ [10], raising hope for its use as an active channel in field effect transistors [11].

When compared to bulk Si and Ge, silicane and germanane offer the compelling combination of a quantized thickness and an atomic flatness characteristic of 2D materials. Both assets reduce the scattering mechanisms involved in electrical transport and thus favor a high carrier mobility. However, it has been shown that the hydrogen termination is unstable against thermal treatment [5], calling for more robust functionalization, such as methyl-terminated surfaces, which can restrict the formation of traps and thereby minimize carrier scattering [12,13]. While theoretical calculations predicted that the carrier mobility in methyl-terminated germanane could reach 10^4^ cm^2^·V^−1^·s^−1^ [14], the experimental works on the transport properties of methyl-terminated germanane remain scarce and are limited to ensembles of flakes [15].

Here, we took advantage of multi-probe scanning tunneling microscopy in ultrahigh vacuum (UHV) to characterize the transport properties of individual microcrystallites. Prior to the electrical measurements, our study revealed a different behavior between microcrystallites under the electron irradiation of a scanning electron microscope despite their annealing at 180 °C. While a fraction of the microcrystallites was well resolved, many microcrystallites became charged. Given that germanane is a layered material [16,17], and that layered materials are known to easily intercalate atoms and molecules [18], we first examined the chemical properties of the microcrystallites to identify the origin of the charging effects. By combining energy-dispersive X-ray (EDX), Raman, and cathodoluminescence (CL) spectroscopies, we show that an incomplete dehydration, or a partial oxidation, accounts for the charging of the microcrystallites and predominantly affects the microcrystallites with lateral dimensions exceeding ~5 μm. In contrast, the stable microcrystallites are always free of oxygen, which is suitable for the formation of good electrical contact. Measurements of their four-point probe resistance is consistent with a bulk transport. While the resistance values normalized by the microcrystallite thickness is comparable to those reported for H-passivated germanane flakes [10,19], the methyl-terminated germanane microcrystallites are found to be thermally robust at 280 °C, a temperature more suitable with standard complementary metal oxide semiconductor (CMOS) technological processes.

## 2. Materials and Methods

The methyl-terminated germanane were synthesized as follows: a three-neck round bottom flask was taken into a N_2_ filled glovebox, to which iodomethane and acetonitrile were added. The flask was then connected to a Schlenk line and immersed into liquid nitrogen until the solution was frozen into a solid. CaGe_2_, water, and a stir bar were added to the flask while the contents were frozen. The contents of the flask had a molar ratio (CaGe_2_:iodomethane:water:acetonitrile) of 1:30:10:60. The flask was evacuated and refilled with nitrogen three times, and the methylation proceeded for seven days at room temperature. At this point, the reaction mixture was again frozen by immersing the flask in liquid nitrogen and loaded into a glovebox filled with N_2_. The methyl-terminated germanane was separated using vacuum filtration, washed with acetonitrile, then dried under vacuum on a Schlenk line. The material was finally redispersed in an isopropanol solution.

The structural and chemical analysis of the methyl-terminated germanane microcrystallites considered in this study was reported in Ref. [15]. Transmission electron microscopy (TEM), X-ray diffraction (XRD), Fourier transform infrared (FTIR), and Raman spectroscopies revealed the formation of fine crystallites with a methyl functionalization of the sheets. Recent additional selected area electron diffraction (SAED) in the TEM confirms the structural quality of the flakes and microcrystallites when methyl-terminated germanane is stored in isopropanol (Appendix A).

For this new study, the crystallites were deposited on a native oxide layer at the surface of a *p*-type B-doped Si(111) wafer by drop casting the crystallites from the isopropanol solution. Two scanning electron microscopes (SEM) were used to perform the characterization of the microcrystallites. The accelerating voltage and probe current in both microscopes were set at 5 kV and 100 pA, respectively. The first microscope (Zeiss Gemini) was installed in an ultrahigh vacuum (UHV) system (nanoprobe, Omicron Nanotechnology), with a base pressure lower than 5 × 10^−10^ mbar. It was used to guide the positioning of four tips of a multiple-tip scanning tunneling microscope (STM) on the crystallites. As the use of good electrical contacts is essential to measure the resistance of a microcrystallite, tungsten tips were prepared by an electrochemical etching in NaOH and thoroughly annealed in UHV to remove the thin oxide layer covering the tips. In order to determine the microcrystallite thickness, a STM tip was brought into the tunneling range above the substrate in the vicinity of the microcrystallite. It slowly scanned the surface and, once the edge of the microcrystallite was detected, safely retracted to keep the tunneling current constant. Based on the variation of the piezo tube along the z-direction, the height profile was measured (Appendix A). The STM tip could also be used to manipulate the microcrystallite and flip it around to visualize its morphology. The cross-sectional SEM view allowed the microcrystallite thickness to be measured (Figure 1c1,c2). The thinnest microcrystallite consisted of a hundred nanometers (Appendix A), which corresponds to a stacking of about a hundred layers [18], whereas a large majority of microcrystallite had a thickness of about 800 nm. As for the transport measurements, one of the tips was first brought into electrical contact at the surface of a microcrystallite with the Si substrate grounded. The substrate was then disconnected from the ground and the three other tips were approached to the surface of the microcrystallite in tunneling mode, so that the current flowed through the microcrystallite only. The final approach was monitored with the tunneling current. Stable electrical contacts were obtained when the current saturated, yielding electrical resistances between the first tip and one of the other contacted tips in the range 1–10 MΩ. The tips were positioned with an in-line four-point geometry. Injection of the current *I* through the outer tips and measurement of the voltage drop *V* between the two inner tips provided access to the four-point resistance *R*_4p_ = *V*/*I*. The four-point resistance *R*_4p_ is independent of the contact resistance [20], but it is important to minimize the drift of the piezo tube and maintain steady contacts during the transport measurements; otherwise, the *V*(*I*) characteristics might deviate from a straight line (Appendix A).

The second SEM was a ZEISS ULTRA 55 scanning electron microscope combined with a Quanta 200/Flash 4010 EDS detector (Bruker, Billerica, MA, USA) or a CLUE system (Horiba, Kyoto, Japan). The fully automatized compact optical spectroscopy module (R-CLUE) with a retractable parabolic mirror offers colocalized Raman, cathodoluminescence (CL), and photoluminescence (PL) imaging, as well as spectroscopy characterization of individual microcrystallites. The Raman spectroscopy measurements were carried out using an iHR320 spectrometer with a laser operating at 532 nm wavelength as an excitation source. The spectra were calibrated by setting the silicon phonon mode at 520.5 cm^−1^.

## 3. Results

The methyl-terminated germanane microcrystallites were first examined with the UHV-SEM of the nanoprobe system. Prior to their observation, they were annealed for 3 h in the UHV preparation chamber. Indeed, the first reflection in the XRD pattern, which corresponds to the interlayer distance, was measured at 2*θ* = 7.9° [15], and a comparison of this reflection with the literature suggests a hydration of the microcrystallites [18]. Although water is typically desorbed at 120 °C in UHV, the annealing was performed at a higher temperature of 185 °C to increase the efficiency of the desorption process as water molecules are intercalated between adjacent GeCH_3_ layers deep into the microscrystallites. Figure 1 shows a selection of microcrystallites with lateral dimensions in the micrometer-scale range. Surprisingly, two types of behavior were found under the electron beam: some microcrystallites appear bright and are well resolved (Figure 1a), whereas the contrast of other microcrystallites show strong fluctuations (Figure 1b). We note that the stable microcrystallites are not altered by a longer exposure to the electron beam. In contrast, the fuzziness of the blurry microcrystallites vanishes upon manipulation with a metal STM tip, as illustrated by the comparison shown in Figure 1c,d. As the STM tip was grounded, we attribute the improved stability of the SEM image to a discharging of the microcrystallite through the electrical contact with the STM tip. A statistical analysis of the occurrence of instabilities in SEM images was performed on 75 microcrystallites, where both types of microcrystallites were observed side by side (Appendix A), revealing that the largest microcrystallites are more likely to become charged (Figure 1d).

To understand the origin of the charging effects, the samples were transferred to a SEM capable of performing EDX spectroscopy. Despite a brief exposure to air, the same behaviors were found. As shown in Figure 2a, the microcrystallite to the right appears stable, whereas three other microcrystallites, delineated by blue arrows, exhibit charging, although the accelerating voltage and the beam current were minimized to 1 kV and 100 pA, respectively. EDX revealed that these microcrystallites contained a large amount of oxygen in contrast with the absence of oxygen for the well resolved microcrystallite (Figure 2b,c). As the detection of oxygen can be caused by the oxidation of germanium or the presence of water molecules still trapped in the microcrystallites, further analyses of the microcrystallites were performed with Raman spectroscopy implemented in the SEM setup. Figure 2d shows three typical spectra, acquired on a well-resolved microcrystallite and on two microcrystallites prone to charging effects, respectively. The well-resolved microcrystallite shows a strong peak at 299 cm^−1^ and a small and broad band in the range 525–635 cm^−1^, with both features also present when the analysis was quickly performed in air (Appendix A), consistent with previous results [15]. We attribute the first peak to the E_2g_ doubly degenerated longitudinal and transversal modes of the Ge–Ge bonds in germanane [6]. The second one arises from two contributions: the second-order phonon modes, similar to what is observed in bulk Ge [21], and the excitation of a Ge–C vibrational mode, a signature of the methyl-terminated germanane. This mode was measured at 594 cm^−1^ in Ref. [15] and is known to occur at 573 cm^−1^ in Fourier transform infrared spectroscopy [12,22].

As to the charged microcrystallites, two distinct spectra were obtained. In the first case (bottom), the spectrum shows a strong shift of the main peak to 289 cm^−1^, with a broad shoulder towards lower wave numbers. This shoulder can be caused either by the presence of amorphous Ge or by the oxidation of the microcrystallite. As a small peak was also measured at 440 cm^−1^, characteristic of the vibrational modes of GeO_2_ [23,24,25], we can identify these microcrystallites as oxidized in agreement with the EDX analysis. The second type of spectra resembles the one of the well-resolved microcrystallites, albeit in a shift of the main peak to 293 cm^−1^, and the occurrence of an ill-defined band around 167 cm^−1^. An aging analysis of microcrystallites in air (Appendix A), which shows a similar magnitude in the shift of the main Raman peak upon a three-week exposure to ambient air, suggests a hydration of the flakes. Regarding the low-frequency band, it could be assigned to the A_1g_ out-of-plane transversal optical mode of Ge–Ge bonds. This mode is expected at a frequency below the E_2g_ mode in germanane [26], and due to the heavier mass of the methyl group compared to hydrogen, could occur at an even lower frequency than the one measured at 228 cm^−1^ in H-terminated germanane [6]. However, water is known to produce a strong Raman peak in this low-frequency region [27]. This is also true for H_2_O molecules under nano-scale confinement [28]. Comparison to Raman spectroscopy performed in air, where a small peak is measured at 162 cm^−1^ (Appendix A), supports this hypothesis. Moreover, a very weak CL signal, centered at an energy of 1.82 eV (Figure 2e), was measured in contrast to the other microcrystallites, where no luminescence was detected. This agrees with the photoluminescence observed in air (Appendix A), indicating that water molecules are still present in the microcrystallites.

Therefore, three types of microcrystallites are found: oxygen-free microcrystallites which show a high stability under the electron beam; hydrated microcrystallites; and oxidized microcrystallites. We tested a longer annealing time at 180 °C for further improvement of the hydrated microcrystallites, but did not notice any change in the probability of finding stable microcrystallites with SEM. In order to understand why intercalated water molecules cannot be desorbed from the hydrated microcrystallites, Raman mapping of crystallites was performed in air. This revealed a variation of the main Raman peak as a function of spatial location within a crystallite: 301.6 cm^−1^ at the center of the crystallite and 304.1 cm^−1^ at the edge of the crystallite (Appendix A). We attribute this spatial variation of the main Raman peak to the chemical environment of the germanane sheets. Indeed, the energy of the E_2g_ mode depends on the nature of the ligands, with the methyl group yielding the highest energy in comparison with H, CH_2_CH=CH_2_, or CH_2_OCH_3_ functionalization [29]. This also depends on the ratio between the number of H and CH_3_ when the relative stoichiometry of both ligands changes. As a result, we suspect the hydrated microcrystallites to be partially functionalized with the methyl groups, making the complete desorption of intercalated molecules more difficult as water molecules can react with poorly passivated sites. As time in air increases, the crystallite further interacts with ambient humidity, which red shifts the E_2g_ mode (Appendix A), further supporting an incomplete methyl passivation.

Based on all of these observations, we suspect a higher degree of polycrystallinity in the largest microscrystallites, with grain boundaries facilitating the ingress of solvent and water molecules during their long storage in isopropanol. As a result, these molecules can further react with germanium atoms through the numerous defects at the grain boundaries, accounting for the detection of oxygen in the EDX experiments. Interestingly, when a charged microcrystallite is manipulated with the STM tips, it can be cleaved (Figure 3). Following the cleavage, the smaller crystallites do not exhibit a fuzzy contrast in comparison with the biggest one, which is still glittering under electron irradiation. The improved stability of the contrast observed on the small crystallites points to a higher structural quality.

The microcrystallites were electrically characterized with multiple-tip STM. Charged microcrystallites were investigated, but it was not possible to obtain stable electrical contacts. This situation is caused by the difficulty of keeping the charged microcrystallites immobile, when polarized tips are approached or contacted. Indeed, microcrystallites can move because of the electrostatic repulsion induced by the polarized STM tips (Appendix A). Moreover, when contact is achieved, due to their partial oxidation, the insulating character of these microcrystallites results in highly resistive contacts which preclude the injection of current into the microcrystallites. Conversely, on the well-resolved microcrystallites (Figure 4a,c,g), the *V*(*I*) characteristics were rather linear, as shown in Figure 4c. Analysis of the resistance as a function of the tip separation revealed two types of behaviors among these microcrystallites. For microcrystallites, where the separation of the tips is in the range of the microcrystallite thickness, the resistance increases with the tip spacing. For example, the microcrystallite seen in Figure 4a has an electrical resistance that linearly increases for more than 4 kΩ, when the tip separation varies between 0.7 μm and 2.5 μm (Figure 4d). Due to a crystallite thickness which is smaller than the outer tip separation, the current distribution is compressed at the bottom of the crystallite, raising its electrical resistance (Figure 4b). Conversely, for thicker microcrystallites where the tip separation is smaller than the microcrystallite thickness, the current predominantly flows near the surface. Hence, the resistance is inversely proportional to the distance between the tips. Such an example is illustrated in Figure 4f for the microcrystallite observed in Figure 4e.

While strikingly different, both behaviors are consistent with a three-dimensional transport [20]. At small tip distances, despite the finite size of the microcrystallite, the four-point resistance verifies Ohm’s law for a homogeneous and isotropic semi-infinite three-dimensional resistive material, *R_4p_* = *ρ*_3D_/2π*d*, with *ρ*_3D_ and *d* as the bulk resistivity of the microcrystallite and the tip separation, respectively. At larger distances, the four-point resistance verifies the relationship *R_4p_* = *ρ*_3D_*d*/*S*, where *S* is the cross-section of the microcrystallite. Estimating a 2.5 μm^2^ cross-section for the microcrystallite in Figure 4a and fitting the data points of Figure 4b,d with the second and first relationships, respectively, yields the consistent resistivities of 0.78 Ω·cm and 0.66 Ω·cm. A confirmation of a three-dimensional transport is provided by changing the probe arrangement. Instead of contacting the same top plane, the source tip can be in contact with an edge facet and the potential probes positioned across the microcrystallite as the electrodes for the source. Ground and potential detection are then easily commutable. Such a situation is illustrated in Figure 4g. This might give rise to slight deviation on the *V*(*I*) characteristics due to small changes in the injection of electrons when the tip sourcing of the current is in contact with rough facets. Overall, however, the *V*(*I*) curves yield resistances around 1 kΩ (Figure 4h), which is in agreement with the resistances found before. Although the microcrystallites can be seen as parallel resistors, their manipulation with the STM tips reveal a defective layered morphology, as shown in the SEM side views of Figure 1c. This inhomogeneous structure, with a random presence of dislocations and grain boundaries, renders the estimation of the resistivity of a single methyl-terminated germanane layer impossible.

However, it is interesting to compare the resistance of methyl-terminated germanane microcrystallites with the resistance of H-terminated germanane flakes found in the literature. Although the H-terminated germanane flakes show similar lateral sizes, they were much thinner [9,18]. Hence, we normalized the resistance by multiplying it by the measured thickness of the microcrystallite. Figure 5a summarizes the values of the resistances measured for different microcrystallites with similar electrode spacings, in the range 0.5~3.0 μm. For annealing temperature around 180 °C, the electrical measurements show similar results between the CH_3_-terminated microcrystallites and the H-terminated flakes. At a temperature of 210 °C, the resistance of the H-terminated germanane flakes strongly decreases, whereas an annealing of the methyl-terminated germanane microcrystallites at 280 °C for 12 h does not lead to any significant change in resistance. The stability of the electrical conduction at varying temperatures is supported by the structural analysis of the microcrystallites. As seen in Figure 5b–d, observation of the edge of a microcrystallite thin enough to exhibit a single diffraction pattern does not show any significant modification in the TEM image, as well as in the SAED pattern, highlighting the robustness of the methyl groups. The passivation of Si and Ge surfaces with hydrogen is known to be fragile. The surfaces easily react with water and organics in air, accounting for the lack of PL in H-terminated germanane [12]. The resistivity of H-terminated germanane drops after an annealing at 210 °C, due to hydrogen desorption and the possible transformation of the layers into germanene layers [19]. In contrast, the Ge—C bonds are stronger and more resistant to oxidation. This absence of any electrical modification shows the strong thermal stability of the methyl group, which is consistent with a previous study where the methyl desorption was found to occur at 420 °C [13].

## 4. Conclusions

In summary, methyl-terminated germanane microcrystallites intercalate molecules, water in particular, which can lead to reactive processes and unwanted chemical modifications. While the release of the intercalated water molecules upon annealing in UHV is more efficient for the smaller microcrystallites, a straightforward identification of the chemical quality of the microcrystallite can be easily performed with SEM by identifying the microcrystallites that do not charge under electron irradiation. These microcrystallites show ohmic behavior, which, in contrast to the H-terminated flakes, is found to be stable for annealing temperatures higher than 200 °C, offering stronger reproducibility for future experiments involving germanane flakes in a field effect transistor configuration.

## Figures and Tables

**Figure 1 nanomaterials-12-01128-f001:**
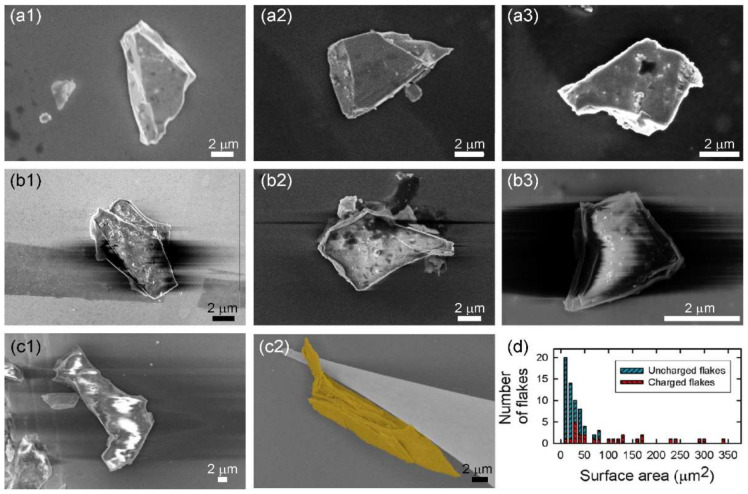
Scanning electron microscopy (SEM) images of typical methyl-terminated germanane microcrystallites drop-casted from an isopropanol solution on a Si surface and annealed at 180 °C in ultrahigh vacuum. (**a1**–**a3**) Examples of microcrystallites which are well resolved with SEM. (**b1**–**b3**) Examples of microcrystallites which become charged under the electron beam. (**c1**,**c2**) Comparison of the SEM image quality upon the microcrystallite manipulation with a STM tip. The crystallite is colorized to be better differentiated from the STM tip. (**d**) Statistical analysis of the occurrence of charging effects as a function of the lateral sizes of the microcrystallite basal plane.

**Figure 2 nanomaterials-12-01128-f002:**
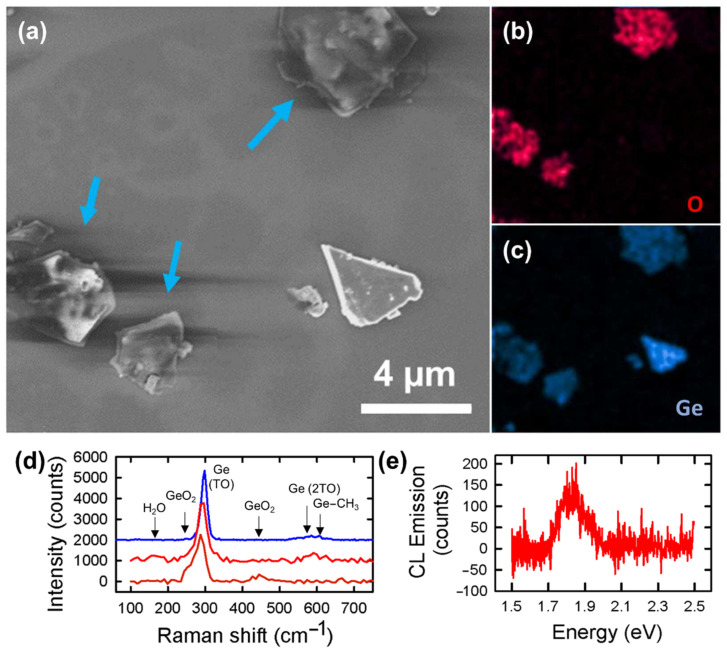
(**a**) SEM image of methyl-terminated germanane and corresponding energy-dispersive X-ray spectroscopy elemental mappings for (**b**) oxygen and (**c**) germanium. Blue arrows delineate microcrystallites that exhibit charging under electron irradiation. (**d**) Raman spectroscopy measured in the SEM setup of oxidized (bottom), hydrated (middle), and dehydrated (top) methyl-terminated germanane microcrystallites. The vertical arrows delineate the positions of vibrational modes of Ge (TO: transverse optical phonon mode), GeO_2_, and H_2_O. (**e**) Cathodoluminescence spectroscopy of a hydrated methyl-terminated germanane microcrystallite.

**Figure 3 nanomaterials-12-01128-f003:**
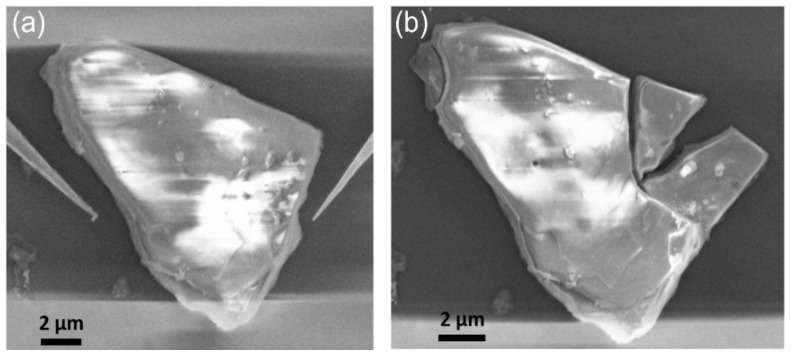
(**a**) SEM of a charged methyl-terminated germanane microcrystallite which is manipulated with two STM tips. (**b**) The right top corner of the microcrystallite was broken, producing two smaller and stable microcrystallites under electron irradiation.

**Figure 4 nanomaterials-12-01128-f004:**
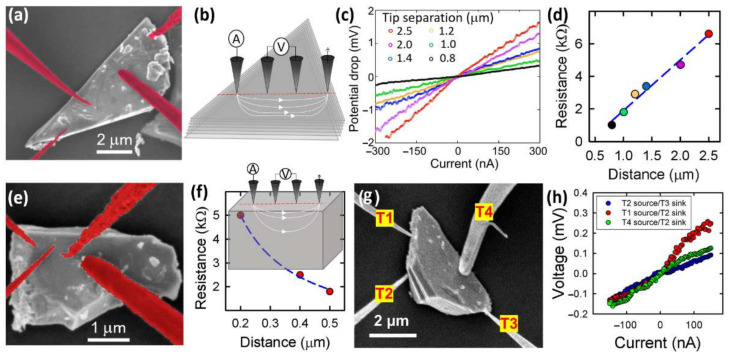
(**a**) SEM image of a methyl-terminated germanane microcrystallite contacted with four STM tips. The STM tips have been colorized to be better differentiated from the crystallite. (**b**) Schematic of the current flow with a compressed current distribution due to the limited thickness of the crystallite. (**c**) *V*(*I*) characteristics measured for six different equidistant tip separations. (**d**) Linear variation of the four-point resistance as a function of the probe separation. The dashed line corresponds to the best fit. (**e**) SEM image of a methyl-terminated germanane microcrystallite contacted with four STM tips and (**f**) related four-point resistance. The dashed line corresponds to the best fit, which is inversely proportional to the equidistant probe spacing. Inset: Schematic illustration of the current flow pattern for a thick microcrystallite. (**g**) SEM image of a methyl-terminated germanane microcrystallite contacted with STM tips from different facets and (**h**) corresponding four-point resistances as a function of the position of the source tip and the grounded tip.

**Figure 5 nanomaterials-12-01128-f005:**
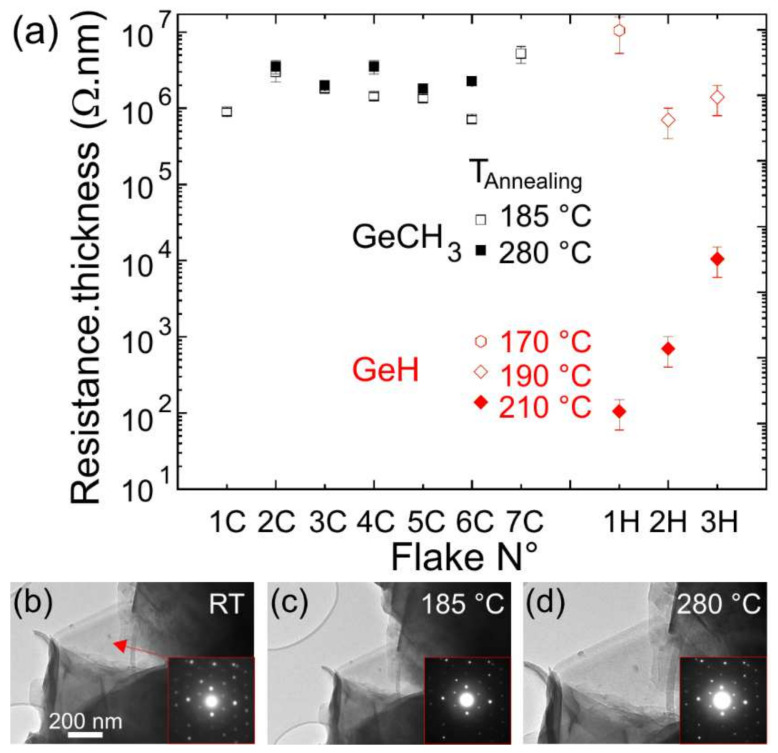
(**a**) Comparison of the product of the four-point resistances to the crystallite thickness for GeCH_3_ (C) microcrystallites and H-passivated (H) flakes, measured as a function of the annealing temperature. The spacing between the electrodes or the tips varies in a range extending from 0.5 to 3.0 μm. The data for the H-passivated flakes were deduced from the experimental results published in Refs. [9,18]. (**b**–**d**) TEM images and SAED patterns of the thin edge of a microcrystallite collected at room temperature (RT), 185 °C and 280 °C, respectively.

## Data Availability

The data presented in this study are available on request from the corresponding author.

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
