# Peer review of "Transport Properties of Methyl-Terminated Germanane Microcrystallites"

_nanomaterials, 2022, doi:10.3390/nano12071128_

Round 1
Reviewer 1 Report
The manuscript submitted by Sciacca et al. presents a study of the characteristics of the individual methyl-terminated germanane flakes with emphasis on the interplay between lateral size, surface functionalization and chemical reactivity. In addition, the four-point resistance of the methylated germanane flakes was measured and compared with other studies in the literature. The particular merit of this paper is that the study of the single flakes is extended to a multitude of nanocrystals allowing to draw general conclusions. Although the scientific content of the manuscript is appropriate for publication on Nanomaterials, the authors should include additional information and respond to the questions before the manuscript can be considered suitable for publication.
Comments to the authors:
- Talking about methylated germanium flakes is a bit misleading for the reader because in the discussion of 2D materials, the term ‘flake’ is usually used for few layer thick nanocrystals. Here instead the minimum thickness is 100 layers according to the authors and the majority of the crystals has a thickness of 800 layers. This referee would therefore prefer ‘crystals’ instead of ‘flakes’.
- On the basis of the synthetic protocol that the authors followed, the produced material should consist almost of fully methylated 2D germanene crystals. Because the crystals are large enough, the authors should prove this by XRD to determine the structure and by (small spot) XPS to gain insight in the chemical composition and the degree of functionalization of the produced material.
- ‘’By combining energy-dispersive X-ray (EDX), Raman and cathodoluminescence (CL) spectroscopies, we show that an incomplete de-hydration or a partial oxidation account for the charging of the flakes, predominantly affecting the flakes with lateral dimensions exceeding ~5 nm’’ (line 58-61). While the authors claimed that incomplete de-hydration could be one of the reasons (among partial oxidation) for the charging effects occurred on the flakes under examination, why they did not prolong the annealing time, or increase the annealing temperature?
- ‘’For the smaller flakes, a full de-hydration can be achieved without any additional chemical transformation, supporting a homogeneous methylation.’’ (line 61-63) How can the full de-hydration of the smaller flakes support the homogeneous methylation?
- ‘’Figure 1 shows a selection of flakes with lateral dimensions in the micrometer-scale range. Surprisingly, two types of behavior were found under the electron beam: some flakes appear bright and are well-resolved (Figure 1a), whereas the contrast of other flakes shows strong fluctuations (Figure 1b)’’ (line 123-127). The authors claimed that the charging effects that occurred on some flakes is a matter of lateral dimensionality and degree of de-hydration upon annealing. Although, Figure 1a and Figure 1b shows flakes with lateral dimensions in the same range, they exhibit different charging states. (well resolved and charging). Can the authors explain this?
- ‘’To understand the origin of the charging effects, the samples were transferred to another SEM able to perform energy-dispersive X-ray (EDX) spectroscopy’’ (line 133-134). Figure 2 a,b,c shows the SEM image and the elemental mapping of different methyl terminated germanane flakes, oxidized, hydrated and de-hydrated. Provided that the samples under examination are methyl terminated germanane crystals, why is the distribution of carbon is not presented in the elemental mapping? What about the degree of methylation? Are the methyl groups are present only on the surface of the flakes or/and in the inner layers? Sonication assisted exfoliation of the methyl-terminated flakes in single or in few layers crystals could give the answer if studied with AFM measurements.
- ‘’Therefore, three types of flakes are found after the annealing in UHV, flakes with a high purity, hydrated flakes and oxidized flakes’’ (line 190-191). It is well known that germanane based materials are easily oxidized on the surface, while the inner layers remain intact. Is there a relation between the type of flake and its thickness?
- “While strikingly different, both behaviors are consistent with a three-dimensional transport[?]”(line 226-227) – I assume a reference is missing here.
- ‘’As the methylation certainly proceeds from the edge of the flakes, our study indicates a higher efficiency of the methylation process for the smaller flakes’’ (line 200-202). How did the authors deduce that?
- Minor issues: The authors should revise the English of the paper because the wording is sometimes not clear: see for example “a multi-physical characterization” (abstract); “It slowly scanned the surface to safely climb on top of the flake” (line 91);
- Typo: “are well-resolved” should be “are well resolved”; “Alt- hough both types of flakes show…” (line 254) Although is not hyphenated correctly (well-known Microsoft bug).
Reviewer 2 Report
The authors report a multi-physical characterization of methyl-terminated germanane flakes as well as the study of their electrical properties. The subject is interesting, however, there are a few points that should be addressed and discussed in the manuscript. Hence, I suggest some minor revision of the manuscript based on the following comments/suggestions:
- The authors report the characterization of single methyl-germanane flakes. However, in the main text they claim that the thinnest flakes consist of a hundred nanometers. In addition, they report an aging analysis as well as AFM and SEM studies (Supplementary Material) of single flakes but from cross-sections (height profiles) the thickness is > 20nm. In the same context, they claim the study/characterization of single methyl-germanane flakes (e.g., line 87, 113), but from the Figures o the manuscript the flakes are much thicker. Please explain this inconsistency.
- In the introduction, the authors mention “Both silicane and germanane have a reduced theoretical band gap compared to graphane, 2.9 eV and 1.9 eV respectively”. In this context, the authors should include some references where the band gap of germanane has been experimentally calculated to 1.4 eV (https://doi.org/10.1002/anie.202010404) and 1.6 eV (https://doi.org/10.1021/nn4009406)
- The insets in Figure 3 (b) and (d) should be made more obvious since they demonstrate the main principle of the applied technique.
- In figure 4, the y-x axis should be made more recognizable (e.g., Resistivity, Flake Number or Flake No)
- Some abbreviations should be included e.g., CMOS, STM in line 51
- Some spelling and grammatical errors should be carefully corrected. For example, in lines 121, 151, 214, 227.
Reviewer 3 Report
Sciacca, et al. reports multiple characterization strategies to understand the post-annealing physical properties and electrical transport behavior of methyl germanane, a 2D hybrid lattice comprised of sp3-hybridized 2D germanium sheets terminated by methyl groups on the surface. In their report, the authors claim that an efficient dehydration of intercalated water molecules was achieved upon annealing flakes with small lateral dimensions. The authors also showed the annealing temperature-dependence of the resistivity of the methyl germanane flakes.
The following are inconsistencies that the reviewer found on this manuscript:
The authors claim that methyl germanane crystallites annealed at 180C are stable and maintained crystallinity
-The only probe that was shown to establish this was local micro-Raman measurements. The authors should consider using complementary techniques to establish the stability of methyl germanane flakes upon annealing. This is especially crucial for single crystal studies that the authors are basing their conclusions from. At least an electron diffraction or HRTEM micrographs should be shown before and after annealing of the samples.
The authors claim that water is intercalated between the layers and that this can be removed via annealing
-First, the authors did not show the necessary characterization to establish the identity and crystallinity of the methyl germanane samples that they synthesized. As such, the degree of crystallinity, methyl surface coverage, and termination of the Ge atoms are not known from the samples that they are using for subsequent measurements. The completeness of the conversion reaction was also not demonstrated which will cause uncertainty in subsequent measurements as unreacted CaGe2 can react with air to form layered Ge terminated with -H or -OH.
-While this is possible, the authors did not show sufficient characterization to establish that it is, indeed, water that is being intercalated between the layers. The authors also solely relied only on Raman to establish the structure of these layers. Their specific claims can only be verified by vibrational isotope studies, complementary FTIR measurements, and electron diffraction/HRTEM studies.
The authors claim that the annealed flakes show Ohmic behavior
-I-V data shown in Fig. 3f show three I-V curves with varying non-linear behavior, one of which even showing a drastic deviation from Ohmic-like behavior. To establish this, the authors should show statistically significant I-V curves showing linear behavior.
-The authors measured the electrical transport behavior of the methyl germanene flakes inside an SEM using STM probes but did not establish how the ultrahigh vacuum and electron beam influences the structural stability and integrity of the crystallite flakes especially since charging has been observed. Also, the authors did not define a clear metric and indicator on which flakes are exhibiting the charging and to which extent these are charging.
While some of the results are intriguing, the authors did not satisfactorily and sufficiently established multiple claims presented in the paper. Control experiments, especially those that are central to the major conclusions of the paper, are also missing. On these grounds, the reviewer recommends the rejection of this paper from publication.
Round 2
Reviewer 1 Report
The authors have replied to all questions and comments in a satisfactory manner and the paper is recommended for publication in its revised form.
Author Response
We thank the reviewer for his/her positive appraisal of our work.
Reviewer 3 Report
While changes were made in the manuscript, there are still considerable issues that are apparent in the revised form of the manuscript. The following are comments to the response (numbers correspond to points raised):
(1) The authors cite a previous publication for the structural characterization of the methyl germanane crystals under study but did not make clear whether these are the same samples that were used in this study. If they were, this is still problematic as the authors still failed to establish that these samples remained stable and are in the form they assume so especially with the significant time difference between their previous reports. This was not mentioned in any way in the manuscript.
(2) The completeness of the reaction was established using HAADF/EDS. While this addresses the question about the completeness of the reaction, the authors miss the important point raised which is the specific content of H2O and methyl/-OH coverage in the sample. Without any FT-IR study, it is challenging to delineate the presence of oxygen (which authors used as an indicator) to establish the chemical identity of the contributing species.
(3) While the PL is one of the indicators for water intercalation in methyl germanane, it cannot be discounted that other factors can also contribute to the PL energy. As such, the PL should be correlated to other characterization techniques (e.g. XRD and FT-IR) that the cited paper performed (ref. 18) completely and in conjunction with the cathodoluminescence data to establish this claim. This is why Points 1 and 2 were raised.
(4) While the authors revised the wording, it is still ambiguous whether the I-V behavior is, indeed, Ohmic. The distance-dependent curves still show non-linearity and I-V curves were only shown for two crystallites (both of which showing some form of non-linearity). Also, and more importantly, these non-linearities are already apparent even in the small mV range shown in the measurements (and will be drastically magnified more non-ohmic in higher voltages). The authors also did not discuss where this non-linearity is arising from and how the resistance values were derived given the non-linear curves. Some form of statistical signatures is crucial for these claims to be made.
(5) A clear metric (or statistical demonstration) was recommended in the earlier review. While the "fuzziness" indicate some form of charging this was not statistically presented in the manuscript, especially since the study is focused on local, single microcrystallite, measurements.
